# Alignment of the Safety Assessment Method with New Zealand Legislative Responsibilities

**Dirk J. Pons** 

Department of Mechanical Engineering, University of Canterbury, Christchurch 8140, Canterbury, New Zealand; dirk.pons@canterbury.ac.nz

**Abstract:** Need—National legislative health and safety (H&S) frameworks impose requirements but grant self-management to organisations. Consequently variability arises in management systems, and some organisations struggle to achieve successful implementation. The risk assessment process is key to the H&S management system, and could benefit from greater consistency and better external alignment with the legislative framework of the jurisdiction. Approach—The harm categories in the New Zealand (NZ) Act were adapted into a consequence scale. A non-linear scale was developed for the consequence axis to represent the disproportional nature of catastrophic harm outcomes compared to minor injuries. A hazard assessment process was devised based on systems engineering methods. Organisational decision-criteria were derived from the communications requirement in the Act, and these thresholds linked to expected treatments. Originality—A method is providing for aligning risk assessments with a national legislative framework, and integrating the technical aspects of risk assessment with the management processes. The approach also more explicitly includes recovery actions in contrast to existing methods where prevention dominates. Regarding the management aspects, it shows how thresholds may be defined relative to the legislation, to give clear expectations regarding treatment and internal communication, thereby assisting executives ('officers' in terms of the NZ Act) meet their duties.

**Keywords:** legal duty; organisational behavior; external alignment

## 1. Introduction

The assessment of occupational health and safety (OSH) hazards, as commonly applied in the workplace, typically comprises the tabular evaluation of consequences and their likelihoods, for various risks. The resulting table of hazards and the solutions (which are variously called treatments, remedies, or mitigation) is called a risk register. Optionally, it may also include residual risks, i.e., a reassessment of the risk assuming the proposed treatments are effective.

Risks are possible future events that might happen, and in the safety context they are invariably negative. The term 'risk' has variable usage, and can either mean the possibility of an event, or the combination of consequences and the likelihood of those consequences.

More generally, risks have the potential to give rise to future consequences, which may be negative (threats or hazards) or (positive) opportunities, and this is the perspective taken by the risk management standard ISO 31000 [1]. The risk assessment used in health and safety (H&S) is a simplified version of risk assessment that only looks at the threat component.

In the H&S application the assessment is consistent with the ISO 31000 approach of partitioning the risk into consequence and likelihood, i.e., those are considered orthogonal. The processes are: identify risks, analyse risks; evaluate risks (quantitative and qualitative methods are available); and treat risks. Those who are interested in a fuller discussion of the treatment of risk, including other options that may be useful at design-time, are referred to the risk management literature. Suitable

starting points for risk management generally are ISO 31000 [1], HB436 [2], and for technology risks in particular AS3931 [3] or the equivalent international standards.

However there are a number of problems with the risk assessment approach as applied to H&S. The methods tend not to be aligned to the legislative framework of the jurisdiction, at least in New Zealand (NZ) which is the region under examination. Furthermore, there is no consistency in practice regarding the scales used in the risk assessments. What follows relates specifically to the NZ situation, though some of the principles are believed to have wider applicability.

## 2. Background to Health and Safety Legislation in NZ

### 2.1. Health and Safety Legislation in NZ

In the NZ case, the relevant legislation is the Health and Safety at Work Act (2015) [4]. As the date shows, the Act is relatively recent. Moreover, it introduced a radically different legislative intent towards safety. The initiating event for the change in legislation was the Pike River mine catastrophe [5]. The resulting investigation into the mining catastrophe [6] identified multiple weaknesses in the legislative landscape prevailing at the time of the accident. For one, the legislation obligated organisations to eliminate/minimise/isolate hazards for their staff. The NZ approach to H&S was, and still is, consistent with the principle that legislation should 'stipulate the duties of those with primary responsibility for OSH measures in general terms, rather than to attempt to regulate a multitude of hazards in minute detail [7]. A legislative system comprises regulations in addition to laws, and the mining industry operated under considerable self-regulation. The mine company had the freedom to make decisions with significant adverse safety implications. For example: they proceeded to put the main ventilation fan inside the mine itself, despite the misgivings of the mine inspector [6]; they defined for themselves a 'non-restricted' zone within the gassy coal seam, where they placed electrical apparatus, and made this decision based on engineering convenience [5]. The investigation found no evidence of any risk assessment prior to placing the electrical services in the coal [6]. The safety systems were ad hoc, staff from subcontractor firms were in the mine, sometimes with inadequate or no safety induction: "Its health and safety systems were inadequate" [6]. Hence the self-management of H&S was ineffective and safety was compromised.

It was possible for executives to avoid the legislative duties if they could show, as in this case, that they were uninformed about risks: "the board of directors did not ensure that health and safety was being properly managed and the executive managers did not properly assess [this]" [6], despite the numerous prior warning signs that methane was regularly at explosive levels. Fundamentally, the managers did not manage the risks competently.

Consequently, the law was changed, resulting in the current act [4], which placed a much greater duty of care on executive officers ('officers') to inform themselves of H&S risks and resource the treatments. It also changed the definition of workers to include anyone at the workplace, irrespective of their employment contract or remuneration. It also created a joint responsibility for H&S for all organisations that had workers at the workplace, as opposed to partitioning the duty. The NZ act created a requirement on executives to show due diligence for H&S in a way complementary to their fiduciary duties under the company act. (For a wider analysis of the effect of catastrophes on the improvement of process safety, see [8]).

### 2.2. Intent of Legislative Frameworks for Self-Management—The International Context

The intent of legislative frameworks is that organisations implement 'voluntary arrangements to strengthen compliance with regulations and standards leading to continual improvement in OSH performance' [9]. In practice, this means that organisations are left to develop their own H&S management systems, as evident in ISO 45001 and the ILO standards [10]. (For a comparison of the prescriptive vs. flexible approach to legislation see [11]).

However, to some extent that appears to be a naive premise. As evident in the Pike River mine disaster, even in developed countries the self-management can fail [6]. A similar observation has been made about the Canadian mining sector specifically the need for audit [12], and for inspection in the United States dam sector [13].

There is also a case to be made that smaller organisations, and organisations in developing nations, struggle to successfully implement self-management. Organisations with less access to professional resources may find it difficult to create such systems. There is evidence from Nigeria that the lack of specificity in H&S legislation adversely affects construction safety [14,15]. Weaknesses in clarity and structure have also been identified in the United Kingdom H&S law [16]. Likewise, the Australian experience has been that small and medium enterprises have difficulties in developing specific workplace practices from the abstract and non-prescriptive legislation [17]. Results from Italy also show that smaller firms are less effective than larger ones at reducing accident rates [18]. Likewise, the Malaysian experience has been that H&S management systems are often difficult to implement to a level that meets the legislation [19]. This has also been found to be the experience in Malawi [20]. It would appear that developing countries generally struggle to develop the systems necessary to give effect to legislative objectives. Presumably, part of this is the lesser availability of H&S expertise to smaller firms, which was specifically identified as the case in Ghana [21]. Related to this is the need to actively develop a professional body of health and safety experts, and even in developed nations this is something that requires deliberate work and support [22].

The case has been made that flexibility in H&S legislation creates greater liability risk in the aviation sector, and that organisations are tempted to respond to this by suppressing the internal reporting of safety risks [23]. Hence, the quality of safety reporting becomes a key determinant in the effectiveness of H&S management systems. The older NZ legislation created just such a perverse incentive for executives to maintain their ignorance of risks within their organisations. There can be other perverse gaps between that which the law intends and what organisations implement. In Australia, there are greater H&S responsibilities for larger construction projects, where size is measured in financial terms, but there is evidence that the monetary threshold is not ideal, e.g., contracts may split to avoid the requirements [24].

*2.3. Challenges to Management Systems*

Many engineering projects involve multinational collaboration, and workforces of diverse cultural origins. Challenges to H&S management systems have been observed in large construction projects with multinational labour in the United Kingdom, relating to work practices, communication, and culture [25].

There are other shared areas where nations operate but without sovereign jurisdiction, and there is a need for coordination of safety. Historically the international ocean has been this domain, to which can now be added Earth orbit and other planets. Safety is one of many considerations in the legislative framework that is evolving for space, currently in an ad-hoc manner [26]. New technologies are another area where common understanding of risks and how to assess them can be valuable, nanotechnology being an example [27].

Food safety is a more down to Earth application for risk assessment [28], and the global interconnectedness of the supply chain means that working towards commonality of these safety processes is advantageous. Again small firms are more at risk, e.g., in China [29]. Better coordination of risk assessment processes and documentation between subcontractors, especially the inclusiveness thereof, has been identified as necessary in the chemical industry [30].

Another area where a common understanding of risk is needed is mental health and psychosocial risk [31]. While mental health is specifically included in some H&S legislation, NZ being an example, the methods for assessing and managing this are only weakly developed if at all. Harmonisation of policy across jurisdictions has been found to have positive effects on psychosocial health in Australia,

with non-harmonised jurisdictions showing reduced communication efficacy and management commitment [32].

## 2.4. Need for More Systematic Methods

The international expectation is that risk assessment should comply with the regulations of the jurisdiction (International Labour Organisation) [9]. The ISO 45001 standard for *Occupational health and safety management systems* likewise identifies the central need for organisations to have processes to identify hazards (Section 6.1.2.1) and assess risks (Section 6.1.2.2) [33]. These documents identify multiple factors to be taken into account. In ISO 45001, these include organisational culture, human factors, history of incidents, emergency situations, other people exposed to the hazard, etc. There is much about *internal alignment,* the development of an H&S management system that takes into account how the organisation conducts its operations. The concept that organisations may also need to ensure *external alignment* with the national legislative framework is weakly developed in all these documents, and absent regarding risk assessment in particular. Instead each organisation develops its own H&S management system and tools, including risk assessment methods.

There is a need for more systematic methods for use as elements within an H&S management system. It is apparent from the literature that there exist many organisations, especially smaller ones and those in developing nations, which lack resources to create management systems ab initio. Furthermore, there is a need to strengthen the communication mechanisms between those doing the risk assessments, and the managers who have the financial budget to support treatment.

Key to this is the need to anticipate, identify, and evaluate hazards and risks [9]. This is central to the operation of a safety and health management system within an organisation [33]. If the risk assessment are not being done, or done to a lower level of rigour, or not being communicated to managers, then treatments may not be resourced and continual improvement fails. Effective use of risk assessment processes showed positive benefits in Finland [34] and a positive correlation with specific preventative treatments [35]. (For a general background to the Finnish H&S legal systems see [36]).

## 2.5. Standardised Methods

Standardised safety management methods provide organisations with some assurance against liability. Liability in H&S arises even if the law does not change, because of the development of precedents [37], which invalidate treatments that once were deemed sufficient. Hence there is a need for methods that evaluate compliance, and some recent developments have emerged in that direction (e.g., [38,39]).

When the legislative framework changes abruptly, as in NZ, then there is a methodological shock that propagates through industry, as they seek to understand the new implications for their operations, and realign with those requirements. It is to be expected that the release of ISO 45001 may cause nations to review their laws, in which case this type of readjustment may become more common: there is some evidence that readjustment is already the case in Romania [40].

## 2.6. Specific Needs for New Zealand

In the case of NZ the Act implicitly expects practitioners to conduct hazard assessments, but does not specify the methods. Instead the methods are provided in guidance documents and examples provided by the regulator (Worksafe). These show a semi-qualitative risk-assessment method whereby qualitative estimates of consequence and likelihood are converted to ordered scales. In one publication, Worksafe proposed the use of simple linear scales for consequence and likelihood, e.g.,:

'Insignificant—no injuries,
Moderate—first aid and/or medical treatment,
Major—extensive injuries,
Catastrophic—fatalities' [41].

These scales are simple linear ones to which numbers are allocated (typically ranging from 1 to 5). Of itself, this is not necessarily an issue, because there are often insufficient data on which to make a more precise determination. Also, the method is intended to be applied by workers and team-leaders, or at least comprehended by them, hence a simple system is advantageous.

Another example of the type of scale in widespread use in NZ is:

1. Minor
2. Moderate—first aid required
3. Serious harm occurs
4. Major harm—permanent injury
5. Death—loss of life

Note the variability in the definition of major harm in the two examples above. One defines it as *extensive injuries,* the other as *permanent injury.* It will be appreciated that these are not the same thing, and the extent and severity of these injuries is not specified. Furthermore, neither construct appears in the NZ legislation, hence the external alignment is also missing.

The universal premise of ordered scales is that the increments between the categories are of equal value, but this is not validated. Such scales suffer from methodological invalidity when used in a subsequent mathematical product operation with a similarly constructed likelihood scale, which is invariably how they are used. The issue is the ordinal scale is subsequently being used as an integer one. The reasons why this is problematic are apparent when considering the intervals between the categories. In effect the scale asserts that death is five times worse than a minor incident (such as a scratch that needs a sticking plaster), and that death is another 25% worse than major harm. These are naïve assumptions that do not stand up to scrutiny, however one looks at them. If one applied the perspective of economic rationalism, the cost of a sticking plaster is a few cents, and the financial value of a human life (there are multiple measures) is in the order of a hundred thousand dollars, which is certainly not a 1:5 ratio. Furthermore the death of one person might be a random accident, but the death of many in a catastrophe implies a serious failure of the of H&S management system and may attract fines and punitive consequences beyond the mere proportional increase in the number of deaths. If instead one applied a medical rehabilitation lens, one could measure the diminished quality of life caused by various levels of harm. Quality of life scales do exist, e.g., WHODAS (which measures a person's ability to function in matters of living) [42], and have been applied experimentally to the risk assessment process [43], but do not support simplistic 1 . . . 5 consequence scales. Such scales fail to recognise the non-linear effect of catastrophic accidents.

Providing an organisation uses the same scales throughout, it should result in a consistent measurement of risk. However there is a lack of consistency between organisations—each uses its own scale constructs. Consequently, the same hazards may be assessed differently, and hence given different treatment priorities, between organisations. This is where the problem arises, because H&S legislation tends to make the best practices the standard, and hence organisations may find that they expose themselves to legislative risk.

The regulations define a 'prescribed risk management process' [44], but this is merely the need to identify hazards and apply the hierarchy of hazard control. The actual risk assessment process is not prescribed, and the process goes straight from identifying hazards to treatment without mentioning the types of risk assessment processes that would ordinarily be understood from ISO 31000.

Furthermore the scales are primarily focussed on preventative activities. Indeed that is the perspective taken by the NZ regulator, in common with many other jurisdictions. Thus the prevention of a hazard by elimination or minimisation is the primary objective. This is evident in the hierarchy of hazard treatment which Worksafe uses, for which they use the term 'hierarchy of control measures' [44]. That the focus is prevention is evident in the term 'control' and by scrutiny of the diagram. The concept of recovery mechanisms, to prevent a small accident propagating into a larger catastrophe, is absent (except perhaps tacitly). Worksafe do identify catastrophic hazards, but their definition of

'low frequency, high consequence (e.g., major industrial, workplace or transport incidents such as a large explosion' [45] is focussed on accidents that occur immediately, and ignores the progressive and cascade failures where opportunities to prevent propagation are missed.

For example, the Pike River Mine recorded excessively high methane levels in the days and hours preceding the disaster, which were definitive indicators that the methane risk was out of control. Yet, management ignored that information and continued mining operations that would release further methane. They had previously experienced occasional small localised explosions of methane that were minor and which did not propagate to full conflagrations. The thermodynamic explosion of the mine occurred on 19 November 2010, but the causal path to disaster commenced months before. The real H&S violation at Pike was not so much the final explosion, but the failure to apply recovery mechanisms earlier when it was apparent from the methane readings that the preventative hierarchy of control had failed.

Where catastrophic outcomes can reasonably be foreseen, it is irresponsible to rely only on prevention: thought needs also to be given to recovery. The hazard assessment method is primarily a preventative method, at least in the way it is commonly applied, and is poor at eliciting disaster recovery actions. The bowtie method is much better at soliciting recovery actions that may prevent an accident propagating into a wider disaster. Some of that thinking needs to be included in the H&S risk assessments.

Another issue is that the risk assessment scales bear no resemblance to the definitions in the NZ H&S legislation. This is particularly true of the consequence scale, where legislation may have specific definitions for types of harm, but these are not always carried through to the scales. Consequently there can be a mismatch between the risk assessment scales used by an organisation, and the expectations of legislation. The Act uses a coarse definition of harm, which is notifiable incident, notifiable accident, and death. There are much finer grades of harm that exist, which are important for hazard analysis.

Furthermore, the hazard assessments tend to be ad hoc. The list of risks in the register tends to be in the order in which they were thought up, without a system. Worksafe itself offers no very systematic method for hazard identification [44–46]. Obviously systematic methods do exist, e.g., fault tree analysis, but these seem not to be used. Presumably this is because the purpose of hazard assessments is that they be done by workers rather than safety specialists.

A further deficiency of many hazard assessments is the lack of connectedness of the organisational decision making. The Act created a legal duty for executives to ensure they keep themselves informed of risks. Executives are personally liable if they fail in this duty. In this way, the Act explicitly removed the previous defence of ignorance that was widely used by executives. Hence, there is a need to be more explicit about the thresholds where knowledge of risk is escalated. Existing methods appear not to have a coherent communication strategy and hence executives, at least in NZ and other nations that adopt similar legislation, are potentially exposed to significant personal risk. There is a need for a better way to communicate about risk.

## 3. Method

### 3.1. Purpose

The premise of this paper is that there is potential value to society in being more systematic about H&S management systems. The risk assessment is identified as the key component of an organisational H&S management system. It is suggested that useful improvements would be (a) better alignment to the legislative framework of the jurisdiction ('external alignment'), (b) standardisation of process, especially regarding the scales used. From the perspective of communication theory, the risk assessment might be considered a type of boundary object: an artefact that is passed between people and helps them make sense of the discussion in which they are involved. Hence improving this document has the potential to improve the communication and decision processes.

External alignment is potentially valuable in helping an organisation fulfil its legal obligations, and better understand its exposure to liabilities. It also has the benefit of creating coherent safety terminology throughout organisations, which has the potential to improve communication about risks.

Standardisation of process allows risks to be assessed in consistent ways. This is potentially valuable in giving organisations confidence that their risk assessment process will stand up to external examination. Hence this may reduce liability. It also has the potential to allow risks to be assessed in ways that may be compared across organisations. This may assist benchmarking, and develop a common understanding of risks within industries. Furthermore, standardised processes could potentially be of significant benefit to small organisations and those in developing nations, as they could be provide core components of an H&S management system that could be quickly implemented in a resource-scarce environment.

It is therefore suggested that, in principle, it should be possible to use systems theory to create a basic H&S management system from a set of coherently interacting standardised modules. These could be easier to implement in under-resourced organisations than the customised monolithic systems that well-resourced organisations can afford to implement. The risk assessment process is identified as the key module, and is the focus of the current paper.

Desirable attributes are:

- Provide an instrument which has a clear and unambiguous definition for the various legal categories of harm. This is necessary for consistency of application.
- Accommodate the particular definitions of harm used in the jurisdiction (NZ in this case).
- Accommodate health considerations (chronic, long-term harm).
- Accommodate the non-linearity of catastrophic harm incidents.
- Accommodate the fact that the Act's legal penalties and liabilities increase disproportionately with the scale.
- Define clear expectations regarding treatment. In contrast existing methods leave this to each organisation to set, without guidance.
- Integrate preventative and recovery treatments, by providing a means whereby both may be brought to mind. In contrast existing methods focus overly on prevention at the expense of recovery.
- Offer a systematic way to structure the hazard assessment without overburdening it with the need for specialised knowledge.

The present paper offers solutions towards some of these objectives.

### 3.2. Approach

The approach was to take the harm categories in the Act, and include them explicitly in the consequence scale. As only a few such categories are defined in the Act, it was necessary to interpolate additional categories.

Next, a non-linear quantitative scale was developed for the consequence axis. This, it is proposed, more adequately represents the disproportional nature of serious harm outcomes compared to minor injuries. Also, the scale was extended to include a new category of catastrophe (multiple deaths). This was introduced to elicit disaster recovery actions.

A hazard assessment process was then devised. This is consistent with ISO 31000 [1], but frames the process in a way to encourage (a) systematic consideration of risks, and (b) disaster recovery. Basic principles of system engineering are introduced, by suggesting that hazard analysis is preceded by sketching/describing the system architecture. Several specific methods are offered.

Also, the process recommends that the current state of the technical system be described, especially the existing controls/barriers/procedures that are in place. In this way some of the key thinking behind the bowtie and barrier methods is introduced, without additional burden.

We then considered the organisational decision-criteria. This relates to the legal duty for executives ('officers' in the wording of the Act) to ensure they are kept informed of risks. A set of thresholds were devised. These thresholds were represented both quantitatively (numerical thresholds of risk) and qualitatively (colour regions on a risk map). These thresholds were then linked to the level of treatment response expected. For example the more severe risks could be expected to involve alerting executives, and have treatment in both prevention and recovery plans. We adopted the typically organisational structure used in NZ, which is Board, chief executive officer (CEO), Technical manager, and Team leader.

## 4. Results

### 4.1. A Harm to NZ Legislation

The NZ Health and Safety at Work Act (2015) replaced the concept of 'serious harm' with that of 'notifiable injury or illness' [4]. In this context, notification means towards the regulator, which is a government agency called Worksafe. The definition of notifiable injury or illness is given in Section 23 of the Act, and includes serious types of injury, chemical exposure, or infection necessitating medical treatment. Some types of serious injury are specifically delineated, such as amputation, head injury, eye injury, burns, spinal injury, unconsciousness, and lacerations. Accidental Death is also a notifiable event. The Act was written such that other injury or illness could be added to the list by simple declaration via regulation (as opposed to changing the Act). Accidents that merely require first aid are not considered serious harm or notifiable incidents, unless other factors are involved like zoonoses.

Then the Act defined a second category around near-accidents, and called this 'Notifiable incident'. An incident is when something happened that could have had serious consequences. Generally, and historically, a near-miss had been considered in a somewhat positivist manner. However, the 2015 Act deliberately forced organisations to be more attentive to near accidents. In the New Zealand context, it is important to note that a notifiable incident is one where a person was exposed to serious harm, whether or not serious harm actually occurred. The mere fact that the systems failed to the extent that someone could have been seriously hurt is sufficient.

Specifically, the NZ Act identifies a notifiable incident as an 'unplanned or uncontrolled incident in relation to a workplace that exposes a worker or any other person to a serious risk to that person's health or safety arising from an immediate or imminent exposure to [list]' (Section 24). The list includes chemical leakage, fire, electric shock, structural collapse (multiple types are listed), vessel collision. Provision is made for others to be defined later by regulation.

As this shows, there are two major types of risk in the Act. Yet, as described above, these are not represented in the risk assessment scales. Hence it is necessary to construct a scale that accommodates these two items, and also includes the finer graduations needed for assessment purposes.

The proposed harm scale is:

1. Hazard present but existing controls prevent progression (hazard is controlled by existing treatments and if it materialises the effects are expected to be inconsequential)
2. Incident occurs with no harm (near-accident)
3. Incident occurs and Minor harm results
4. Incident occurs and exposure to serious harm but no actual harm ('notifiable incident')
5. Serious harm Occurs ('notifiable injury or illnesses)
8. Death
10. Catastrophe (recovery systems fail, multiple deaths occur)

It follows the harm categories in the Act, and provides finer categories. It has other features, by design. It emphasises the perspectives of 'existing controls', and 'recovery systems'. It introduces 'catastrophe' as an extreme outcome. It proposes a non-linear quantitative scale, to solve the problem of existing scales where death is often merely one notch above losing a finger.

Furthermore, the scale may be supplemented by recommended actions and legal responsibilities. The full scale is shown in Table 1. The actions shown here are suggestions from the author, and have no formal standing.

**Table 1.** Proposed consequence (harm) scale, aligned with NZ legislation.

| Level | Harm | Actions | Legal Obligation of Managers, Engineers, and Workers |
|---|---|---|---|
| 1 | Hazard present but existing controls prevent progression | Periodically check the efficacy of existing controls. Be alert to the possibility that the addition of new hardware or changed routines may add new hazards. Repeat the hazard assessment periodically. | Be aware of the hazards: ignorance is an offence in its own right. |
| 2 | Incident occurs with no harm (near-accident) | Collect incident reports, review them, and look for trends. Are some risks increasing? Are you seeing new hazards? Are the existing controls effective? Investigate the more significant incidents. | Make sure that a system exists for reporting incidents and that it is actually used, as lack of systems is an offense. |
| 3 | Incident occurs and Minor harm results | As above | As above |
| 4 | Incident occurs and exposure to serious harm but no actual harm ('notifiable incident') | Something slipped: what was it? Interestingly this is just as much notifiable as when serious harm does occur. | |
| 5 | Serious harm Occurs ('notifiable injury or illness') | Notifiable event | |
| 8 | Death | Notifiable event | The organisation has to report the event to the regulator 'immediately' (s51) [4], preserve the site for inspection until authorised by an inspector (s53), and keep records for 5 years (s52). |
| 10 | Catastrophe (recovery systems fail, multiple deaths occur) | Often organisations are slow to recognise the problem, unwilling to admit that a mistake has happened, and tardy in calling for help. Their risk treatment is focussed on prevention, and their limited emergency plans are overwhelmed when catastrophe occurs. They may be paralysed by incapacitation, indecision, or denial as the disaster unfolds. Better recovery systems are needed, but may be difficult to improvise at the time. The fact that catastrophe occurred indicates that the barriers were overwhelmed and the recovery systems failed. Why? | |

There is also inconsistency between organisations regarding the likelihood scale. A quantitative scale based on empirical evidence is best, but is seldom available, so the likelihood scale may be qualitative.

The following likelihood scale is adapted from SAA/SNZ HB436 *Risk management guidelines* [2]:

6. Almost certain

5.  Likely
4.  Possible
3.  Unlikely
2.  Rare
1.  Almost incredible

Note that it is the likelihood of the harm occurring, i.e., takes into account the efficacy of any existing treatment mechanisms, not the likelihood of the root cause arising. The more protective systems already in place, the lower the likelihood of harm. This is why it is important at the scope definition to be clear about what the current state of the system is.

### 4.2. Identification of System Architecture

We propose that risk assessments would benefit from a more explicit categorisation of the threats. If a risk is not identified and included at the early stage, then it will not be included in subsequent analysis and treatment. Creating a system architecture has several potential benefits: It is a check on how well the analyst understands the complexities in the situation. It provides a means to check whether the scope has been covered. Hazards may be inferred directly from the system architecture. It provides an artefact (boundary object) with which to communicate with the rest of the organisation.

We suggest using flowcharts, diagrams, or system representations of the process. Typical categorisations in the work place might be:

1.  Workplace Location: The work may be done by different people in separate physical locations. For example a company printing a newspaper has different risks for the reporters, pre-print layup, printing engineers, and delivery people.
2.  Zones: Another type of location is where there are different technical systems in different physical zones, irrespective of whether there are people there or not. For example, an aircraft fuselage has a cockpit, galley, passenger seating zone, cargo hold, toilets, and avionics bays. Each of these have different types of hazards. A specialised form of this is called Zonal Analysis.
3.  Hardware Sub-System or Type of Equipment: This is similar to above except the analysis is for each piece of plant or equipment. An example is a construction site, and the analysis could be by the type of machine: excavator, roller, dump truck, laser based surveying equipment, etc.
4.  Task: In this case, a worker is performing multiple different types of task. For example, a garden service worker could be mowing lawns (dust, cuts), trimming hedges (hand injury, eyes), spreading compost (Legionnaires disease), or lifting and loading (back injury).
5.  Work-Flow: In most manufacturing plants, there is a sequence of operating procedures that determine the work-flow. In which case, the hazard would be determined for each stage in the work-flow. For example, in an injection moulding firm there are hazards involved with lifting plastic granules into hoppers (falls, working at height dust), heating and drying of feedstock (fire), injection moulding (burns, fluid lancing by blast of hydraulic oil, slipping on floors, fumes, fire), part removal from press (burns, crushing), or deburring (cuts, contact dermatitis, repetitive strain injury).

### 4.3. Decision Criteria

Using the above scales, we identified by inspection of typical outcomes, which the risk thresholds are approximately: 30 or higher—Unacceptable risk; 18 or higher—Urgent treatment; 8 or higher—Consider treatment; 7 or less—No intervention necessary. These thresholds are specific to the scales for harm and likelihood, and would need to be adjusted if the scales changed.

Executives or the board may like to be involved with this as they are personal liable for organisational risk appetite. Is it acceptable or not for a certain number of minor harm injuries to occur?

These thresholds can also be represented in a qualitative form on the risk map, see Figure 1. We have designed it such that the qualitative and quantitative methods give consistent analysis outcomes.

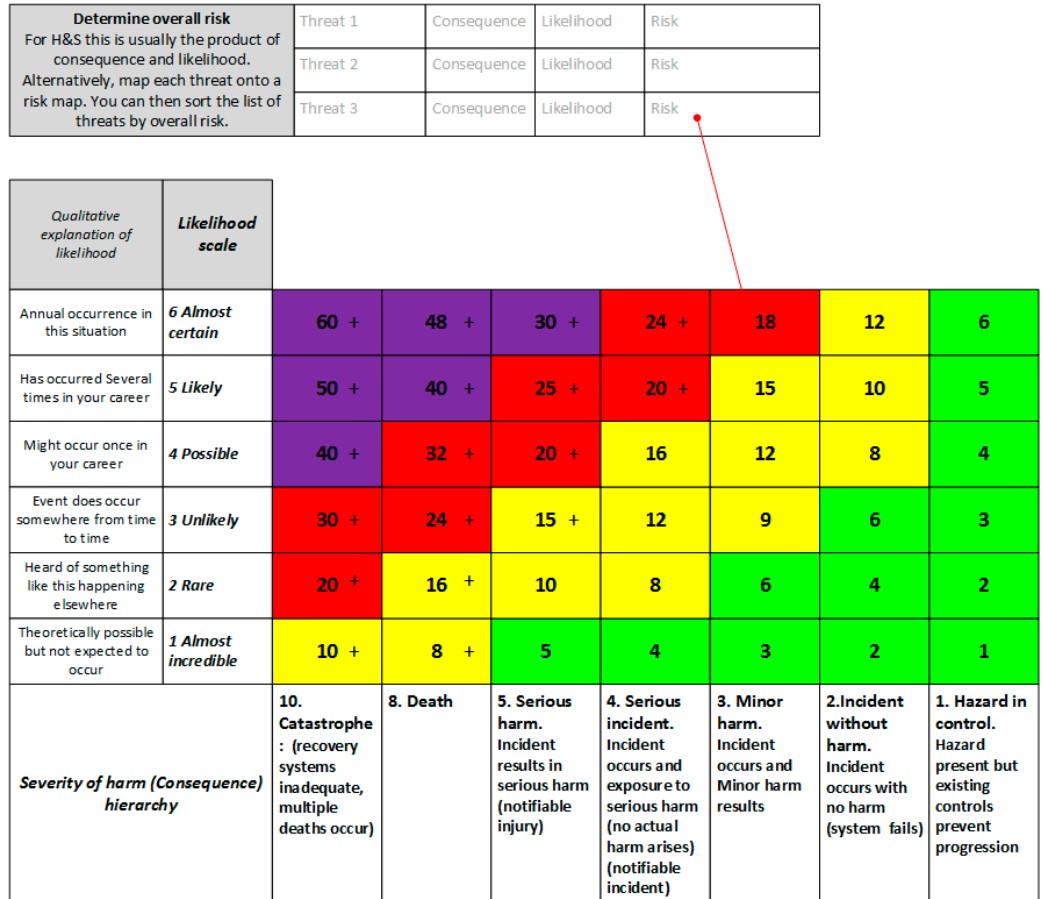

**Figure 1.** Risk map. This provides for both qualitative and quantitative risk assessment. The colours represent risk tolerances per Table 2.

The thresholds and distribution of colour represent the risk appetite of the organisation, hence also where a risk treatment needs to be applied. The thresholds/colours should be set by executives since they will be the ones allocating the resources to the solutions, and carrying the responsibility per the duty of due diligence.

We suggest that a *Principle of Action* be allocated to each magnitude band of risk. For example, for the most severe risk the recommended action is 'Operations to cease until risk reduction is achieved. Ensure provenance of Disaster Recovery mechanisms in addition to Preventative means.' This represents the risk appetite of the organisation.

We also propose another attribute be added to the risk bands, which is to describe where the authority lies for continued operation under these risks. These authority levels correspond to the organisational structure. Nominally this is Board, CEO, Technical manager, Team leader, and these roles could be customised for particular organisations. Finally, we provide a reporting attribute. This indicates who needs to be informed about risks at the various levels. The same roles are used as previously.

Together these attributes are intended to ensure that the organisation has knowledge and control of its risks, while not being overwhelmed by reporting of every small risk. It provides a structured approach to meeting the reporting requirements of the Act. In this way, the risk assessment process provides a mechanism to allocate scarce organisational resources to treat the risks that matter the most. Using the methodology allows the organisation to show that they have approached this resource-allocation decision in a rational manner (have applied due diligence). The results are shown in Table 2. The colour bands in Figure 1 are the same as those in Table 2, and all the content thereof is simply a proof of concept of how the alignment and harmonisation can be achieved. The decision

thresholds have been determined subjectively. If this system was to be implemented in an organisation it is recommended that Table 2 be adapted as needed, and then Figure 1 changed accordingly.

**Table 2.** Decision thresholds with actions and reporting expectations. Colours correspond also to regions on the risk map (see Figure 1).

| Risk, R = C × L | Corresponding Colour in Risk Map | Description | Principle of Action | Authority for Continued Operation | Reporting |
|---|---|---|---|---|---|
| 30 or higher | Purple | Unacceptable risk. | Operations to cease until risk reduction is achieved. Ensure provenance of Disaster Recovery mechanisms in addition to Preventative means. | Board | CEO to advise Board as soon as practicable. |
| 18 or higher | Red | Urgent treatment. | Urgent implementation of treatment required. Operations proceed with caution and ongoing monitoring of risk | CEO | Technical manager to advise CEO as soon as practicable, and report regularly on status of the risk and its treatment. |
| 8 or higher | Yellow | Consider treatment | Implement treatment in a reasonable time period. Operations proceed, with monitoring to detect if risk becomes worse or persistent | Technical manager | Team leader to report periodically to Technical manager on the risk and the progress of the treatment plan. |
| 7 or less | Green | No intervention necessary. | No further treatment required. Operations continue, with ongoing monitoring to check the efficacy of existing controls/barriers/proced Conduct periodic (e.g., annual) re-assessment of the risk. | Team leader | Staff to report periodically to Team leader on the state of this risk. |

*4.4. Treatment Options*

For H&S, the usual treatment options are elimination–isolation–minimisation. This is the conventional hierarchy of hazard control. Elimination is for the engineer to design the hazard out of the system. If that is not practicable, then minimisation is the next solution. Use of personal protective equipment (PPE) is considered the solution of last resort, not the first or only solution.

However, these are only preventative actions. Preventative mechanisms (proactive barriers) prevent the threat from arising or progressing to an incident. We propose that it is also necessary, for the more serious injuries, to consider the recovery mechanisms. Recovery mechanisms (reactive barriers) prevent the undesired state from progressing to further catastrophe. They recover the situation, by breaking the consequential chain. They reduce either the severity of the consequence, or the likelihood of further harm, e.g., emergency response plans.

The latter include recovery actions taken by trained staff (e.g., pilots able to belly-land an aircraft), reserve capability of technical systems (e.g., aircraft fuel tanks that resist rupture), emergency response (e.g., air crew able to evacuate the plane quickly and to do so not into the fire or into the path of the fire trucks), and rescue responses to minimise death (e.g., fire crews to quickly extinguish the fire, quick

access to medical care for critically injured people). These actions happen poorly unless planned in a systematic manner, the Pike River mine disaster being a case in point [5]. The recovery mechanisms are also called barriers to the progression of the accident to a catastrophe. An effective methodology is bowtie analysis.

We therefore propose that the risk treatments need to be more broadly considered than often seems to be the case. We offer Figure 2 as a summary of the various treatment components.

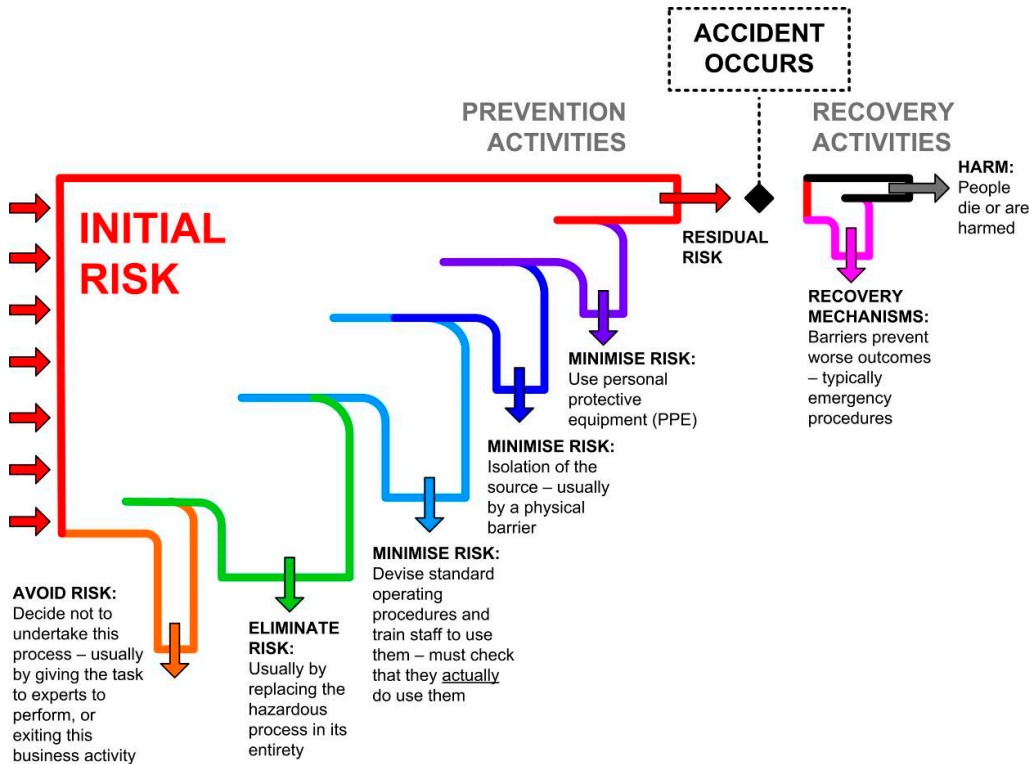

**Figure 2.** Preventative methods include elimination of hazards, minimisation (by isolation, operating procedures, and personal protective equipment). However, there is always a residual risk, even if only because these preventions are imperfect. Once the accident occurs it is only the recovery mechanisms that prevent it progressing to serious harm.

### 4.5. Adjustments to the Method for Safety Assessment

We suggest that in the type of legislative regime as exists in NZ, it is necessary to more explicitly include the organisational interaction with the more technical work streams of risk assessment. We say this because it is the executives who personally carry the financial and criminal risks. Therefore, it makes sense for organisations to arrange their internal operations so as to reduce this risk.

Consequently, we propose some small modifications to the overall process of safety assessment. These are relatively minor and more for emphasis that any substantive change to the ISO 310000 process. We propose there are three work streams for an organisation. The initial management set-up activities are to select appropriate consequence and likelihood scales, and identify the decision criteria. These have been detailed above. Once decided, these do not have to be redefined for each case, but rather they might become part of the standard operating procedures. Experience suggests that organisations do tend to have standard consequence and likelihood scales, and hence this is a call to better align those with the legislative requirements (per 4.1 above), and to more explicitly express the decision criteria (per 4.3 above).

It is proposed that the more technical work stream of risk assessment involves: Identify the system architecture based on current state of plant or operations; Analyse hazards; Evaluate risks; Notify the organisation so that executives are kept informed of risks to which the organisation is exposed; Devise treatments, and Evaluate residual risk. The result of this process is the risk register, an example of which is shown in Table 3. An example of the application is shown in Appendix A.

**Table 3.** Structure of the proposed risk register.

| Architecture level: Work-stream, project phase, hardware category, workstation | Specific hazard | Risks of System in its CURRENT STATE, with Its Existing Controls. | | | | | | Risks of System in its FUTURE STATE after These Treatments. Insert Any New Threats Caused by the Treatments. | | | |
|---|---|---|---|---|---|---|---|---|---|---|---|
| | | Consequence (C), as per 'Severity of harm' scale | Likelihood (L) of that consequence arising | Risk = (C × L) | Treatment? Consider Preventative and Recovery mechanisms. 30 or higher Unacceptable risk. 18 or higher Urgent treatment. 8 or higher Consider treatment 7 or less No intervention necessary. | Action required by who? Resources required? | Monitoring required of efficacy of treatment? | Treated Consequence (C*) | Treated Likelihood (L*) | Residual Risk (C* × L*) | Is this acceptable? What further action is required? |
| | | 1. Hazard occurred | 6 Almost certain | | | | | | | | |
| | | 2. Incident with no harm | 5 Likely | | | | | | | | |
| | | 3. Incident and Minor harm | 4 Possible | | | | | | | | |
| | | 4. Incident and exposure to serious harm | 3 Unlikely | | | | | | | | |
| | | 5. Serious harm Occurs | 2 Rare | | | | | | | | |
| | | 8. Death | 1 Almost incredible | | | | | | | | |
| | | 10. Catastrophe | | | | | | | | | |

In parallel, we anticipate a management work stream which identifies the scope of the assessment in the first place. This might be in response to emergent or insufficiently controlled risks, changes in external imposed regulations, or raised expectations based on how similar organisations are treating their own risks. Importantly, in the eyes of the NZ law, it is also the responsibility of executives to allocate resources to the treatments, keep themselves informed of risk status, and verify the efficacy of treatments.

These proposed work streams and their interactions are shown in Figure 3.

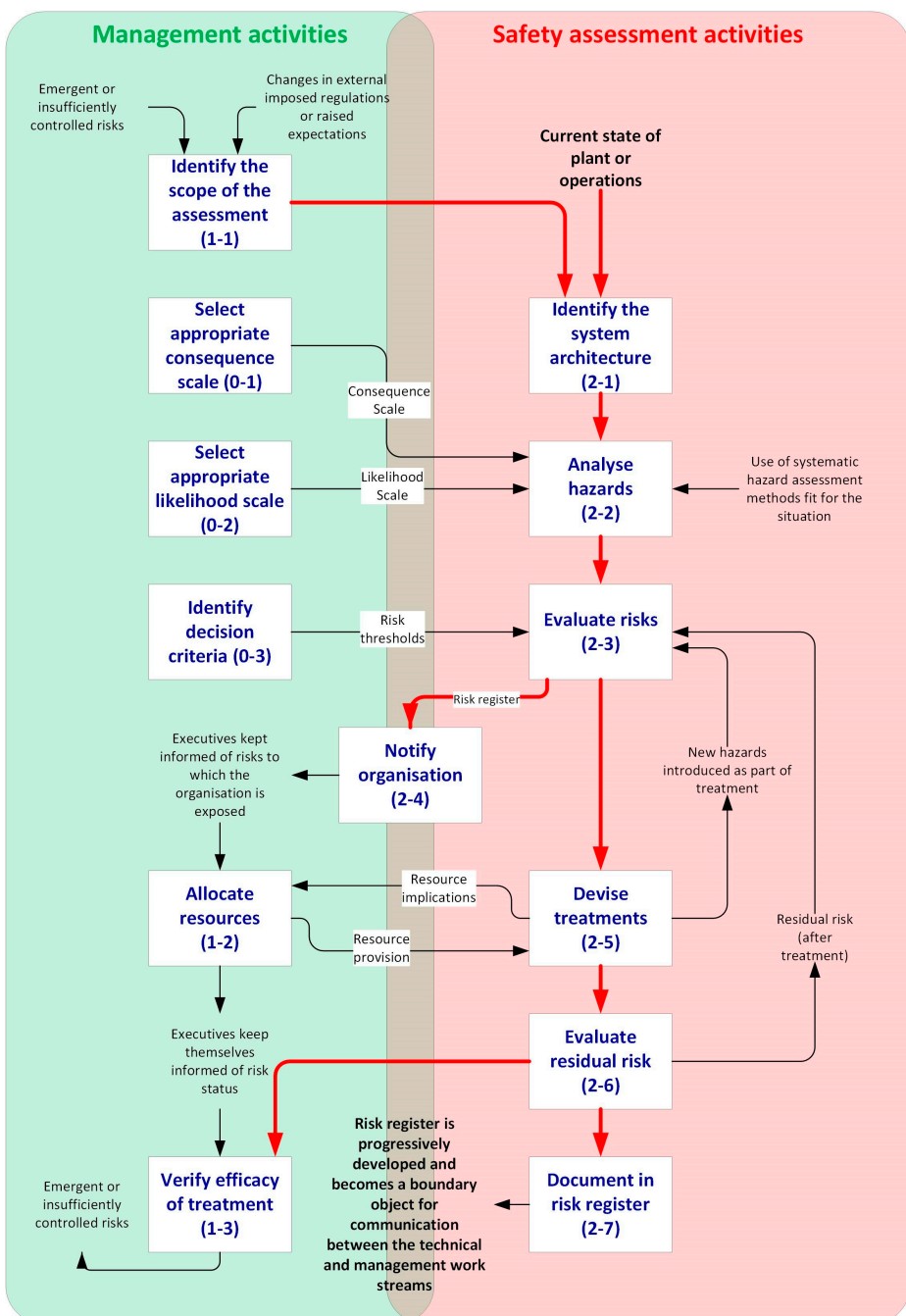

**Figure 3.** Proposed risk assessment process.

## 5. Discussion

### 5.1. Outcomes

This work makes several contributions. The first is developing a method to align safety assessments towards a national legislative framework. The key components to achieving this are to align the harm scale with the national definition of harm, creating a mechanism that links organisational risk appetite to thresholds for management escalation, and providing organisational processes that integrate the technical aspects of risk assessment with the management processes.

A second contribution is more explicitly including elements of systems engineering into the safety assessment process. This is evident in the inclusion of a systems architecture perspective, and in the integrated organisational processes.

A third contribution is more subtle, in the inclusion of recovery thinking in the safety assessment process. This is evident in the non-linearity of the harm scale, the decision criteria that call for disaster recovery mechanisms, and the modification to the conventional hierarchy of hazard control. The collective effect of these is intended to cause the analyst to be more mindful of the need not to stop at mere preventative treatment. This has been achieved without burdening the process with the full barrier/bowtie methodology. Hence, it is a subtle reframing of the process that has been achieved.

*5.2. Implications for Practitioners*

Practitioners are encouraged to make adjustments to their organisational risk assessment processes to achieve better coherency with their national legislative frameworks. As each nation has its own approach to H&S, some customisation of the method is naturally to be expected. The broader principles are expected to be universally valid: Of aligning harm scales to legislative definitions of harm; using systems thinking; incorporating recovery (barrier) thinking into treatments; and creating better communication within the organisation about its risks. The existing safety assessment processes tend to have a large ad hoc component, and the present work offers a mechanism to achieve greater coherence.

*5.3. Limitations of the Work*

The work is conceptual in nature, and hence the main limitation is that the efficacy of the method has not been verified. It has not been tested with users.

The risk assessment method generally has many subjectivities, and the current approach shares many of these. There is the subjectivity of what to include/exclude from the analysis in the first place (which this method attempts to solve by requiring the system architecture to be described), how deeply to identify risks, how to handle complex causal sequences, the ambiguity of what the words in the scales mean, and the uncertainty of the residual risks after the treatments. To a great extent the choice of which analysis method to use, (e.g., risk assessment, fault tree analysis, failure mode and effect analysis, bowtie, etc.) constrains the way the analyst perceives the problem, and hence is another source of subjectivity. In a perfect world all risks would be exactly qualified, but even the quantitative methods often lack precise data and analysts are obliged to use estimates or heuristics, which are also subjective. The point is that subjectivity exists in any method for analysing risks, and is not a disqualifying factor per se. It seems more important to use a method well, and seek to minimise its limitations. The ultimate objective is to continually improve safety outcomes, and the risk assessment is a tool to achieve that rather than an end in itself.

*5.4. Implications for Future Research*

Organisations produce a large number of risk registers. It may be worth exploring how these are communicated inside the organisation, and how they affect wider decision making. We suggest they are a type of boundary object, and hence it may be interesting to examine them from a communication theory perspective. It may also be interesting to examine organisational processes from a change management perspective: how insights about risks might emerge in a bottom up process from the risk assessments themselves, and the top down strategic processes too. It is apparent in many disaster situations that organisations knew about the risks, at least in some part of the organisation, but failed to communicate this or obtain sufficient resources to remedy the situation. We still see many cases where the organisational response to risk failed. Thus, we suggest, the organisational behaviour work streams are just as important as the technical aspects of risk assessment.

Construct validity is something that appears not to have been given much attention in the qualitative risk assessment method. This can be an issue with both consequence and likelihood scales. Do the terms actually mean the same for those providing the input data, as what the analyst assumes? The harm scale proposed here avoids some of these issues by using the same terminology as in the NZ Act. At least the scale is grounded on something that is nationally agreed, which is an improvement.

There remains a residual ambiguity as to how that Act might be interpreted. It could be interesting to see how consistent the semantic interpretations were between people.

## 6. Conclusions

The purpose of the work was to provide a standardised instrument for risk assessment that could be used in the absence of any other more robust methods, and instead of the ad hoc constructs that organisations may create in their attempt to meet the non-specific requirements of the legislative frameworks. While the results given in this paper do not meet all the desirable attributes expressed in the purpose statement, they do perhaps move the field forward towards a more possible future standardised risk assessment instrument for H&S. In terms of the technical aspects of safety assessment, the method provides an instrument that includes the various legal definitions of harm, accommodates the non-linearity of catastrophic harm incidents, provides greater comparability between risk assessments, and proposes the use of systems architecture to help guide the hazard identification process.

Regarding the management aspects, the approach embodies the concepts of organisational risk appetite. It shows how thresholds may be defined to give clear expectations regarding treatment and internal communication, thereby assisting executives ('officers' in terms of the NZ Act) meet their duties. The provision of explicit prompts for communication within the organisation is a positive feature as the literature shows such communication to be important for successful risk management. It also supports the communication requirement in the NZ Act.

The method more explicitly prompts for recovery actions in the treatment plan (in addition to preventative actions) for catastrophic risks.

This paper offers a conceptual method for aligning the safety assessment process to the national H&S legislation, with New Zealand being the situation under examination.

In putting this method forward, the paper does not preclude the use of other methods. This particular risk assessment approach may have merit for use by organisations that desire to improve their management of H&S risks but are small, in developing nations, or in other ways lack resources to develop their own methods.

**Funding:** This research received no external funding.

**Conflicts of Interest:** The author declares no conflict of interest.

## Appendix A. Example Application of the Method

The method is illustrated by a representative study of a product development case. In this hypothetical case, the firm is wanting to develop a new line of power tools. They have decided to start by looking at the latest tool from competitors, and perform a risk assessment before commencing with product design. In this way, they intend to include safety considerations at the earliest stage of the development. The hardware architecture is shown in Figure A1, and the risk register in Table A1.

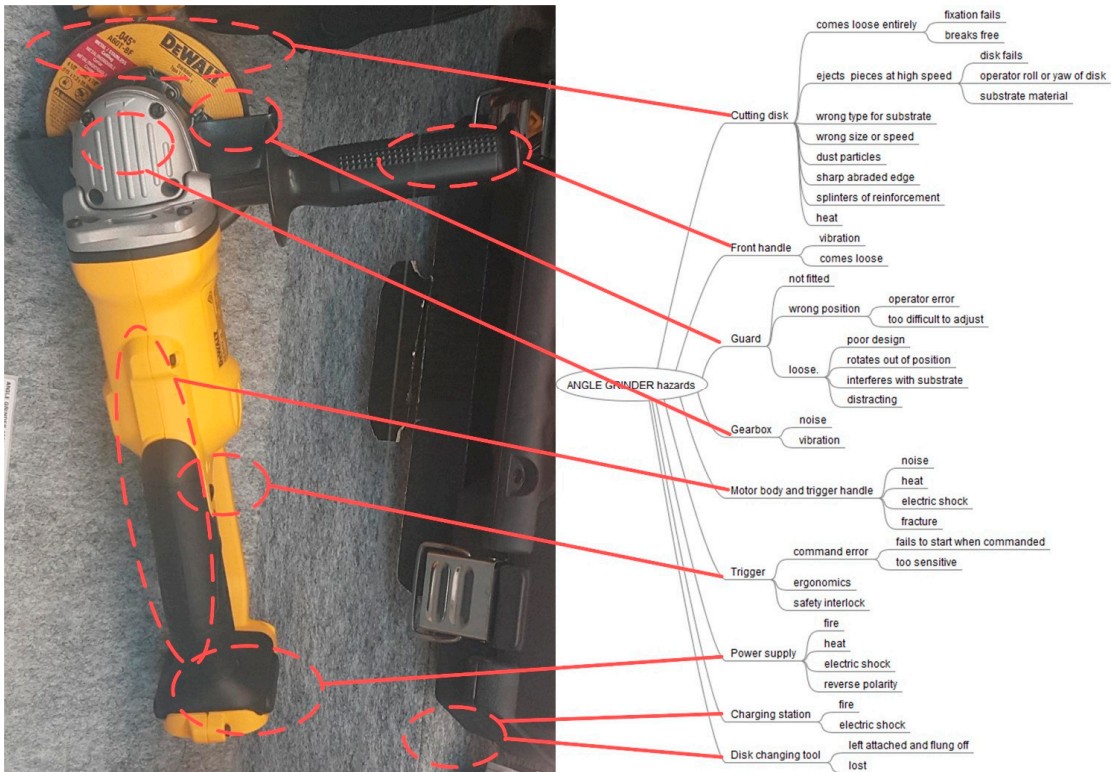

**Figure A1.** Hardware architecture: For this product the functional sub-assemblies provide a useful categorisation of where the hazards emerge. The analysis includes areas where the operator's body parts contact the tool or are in line of sight of ejecta. This example is purely illustrative and does not purport to be accurate or comprehensive. The residual risk after treatment has not been shown here, but the principles are evident in Table 3.

**Table A1.** Risk register for the scenario. This does not include the residual risks.

| Risks of System in its CURRENT STATE, with Its Existing Controls. | | Angle Grinder with Operator Wearing Eye Protection and Hearing Protection | | | |
|---|---|---|---|---|---|
| Architecture level: Work-stream, project phase, hardware category, workstation | Specific hazard | Consequence (C), as per 'Severity of harm' scale | Likelihood (L) of that consequence arising | Risk (C × L) | Treatment? Consider Preventative and Recovery mechanisms. (Controls) |
| | | 1. Hazard occurred | 6 Almost certain | | 30 or higher Unacceptable risk. |
| | | 2. Incident with no harm | 5 Likely | | 18 or higher Urgent treatment. |
| | | 3. Incident and Minor harm | 4 Possible | | 8 or higher Consider treatment |
| | | 4. Incident and exposure to serious harm | 3 Unlikely | | 7 or less No intervention necessary. |
| | | 5. Serious harm Occurs | 2 Rare | | |
| | | 8. Death | 1 Almost incredible | | |
| | | 10. Catastrophe | | | |
| 1 Cutting disk | Fixation fails | 5 | 3 | 15 | Design review to check reliability of fixation between tool and cutting disk. Warning in User instructions: Stop operation if disk develops eccentricity. |
| | Disk breaks free | 5 | 2 | 10 | Manufacturing defect of disk. Warning in User instructions: Do not use cracked or partially broken disks. |
| | Disk fails and breaks up | 5 | 4 | 20 | Poor quality of disk. Warning in User instructions: Do not use cracked or partially broken disks. |
| | operator roll or yaw of disk | 5 | 4 | 20 | Operator error. Warning in User instructions: hold tool steady and avoid bending the disk in the cutting slot |
| | Pieces of substrate material break free | 4 | 2 | 8 | Situational error. Warning in User instructions: No suitable for substrates that have risk of breaking up. |
| | wrong type disk for substrate | 4 | 4 | 16 | Situational error. Warning in User instructions: Use disk appropriate for substrate |
| | wrong size or speed | 3 | 4 | 12 | Operator error. Warning in User instructions: Use disk maximum outer diameter 120 mm. |

Table A1. *Cont.*

| Risks of System in its CURRENT STATE, with Its Existing Controls. | | | Angle Grinder with Operator Wearing Eye Protection and Hearing Protection | | | |
|---|---|---|---|---|---|---|
| | dust particles | 5 | 5 | 25 | | Warning in User instructions: Use respirator protection if dust or fumes are produced. |
| | sharp abraded edge | 3 | 5 | 15 | | Warning in User instructions: Take care when handling cutting disks as these may have sharp edges or splinters. |
| | splinters of reinforcement | 3 | 5 | 15 | | Warning in User instructions: Take care when handling cutting disks as these may have sharp edges or splinters |
| | heat | 3 | 5 | 15 | | Warning in User instructions: Take care when handling cutting disks as these may be hot. |
| 2 Front handle | 2.1 vibration | 3 | 5 | 15 | | Warning in User instructions: Use gloves. |
| | 2.2 comes loose | 4 | 3 | 12 | | Warning in User instructions: Ensure handle is tight before using. |
| | 2.3 Hand slides medially, contacts disk | 5 | 5 | 25 | | Design handle with medial flange to prevent hand migration. |
| 3 Guard fails to protect eyes | 3.1 not fitted | 4 | 3 | 12 | | Warning in User instructions: Not recommended to use without guard fitted. |
| | 3.2.1 wrong position due to operator error | 4 | 3 | 12 | | Warning in User instructions: Ensure guard is positioned to protect against flying debris, and also not interfere with the substrate. |
| | 3.2.2 wrong position because too difficult to adjust | 4 | 5 | 20 | | Design guard to be easy to adjust and reposition securely, without tool. |
| | 3. loose | 4 | 4 | 16 | | Ibid (design solution) |
| | 3.3.3 interferes with substrate | 2 | 6 | 12 | | Ibid (warning to secure) |
| 4 Gearbox | 4.1 noise | 3 | 4 | 12 | | Already prevented with hearing protection |
| | 4.2 vibration | 3 | 4 | 12 | | Ibid (use gloves) |
| 5 Motor body and trigger handle | 5.1 noise | 4 | 4 | 16 | | Ibid (hearing protection) |
| | 5.2 heat | 3 | 2 | 6 | | Ibid (use gloves) |
| | 5.3 electric shock | 5 | 2 | 10 | | |
| | 5.4 fracture | 4 | 3 | 12 | | Design housing for robustness |

<p style="text-align:center">**Table A1.** *Cont.*</p>

| Risks of System in its CURRENT STATE, with Its Existing Controls. | | | | | Angle Grinder with Operator Wearing Eye Protection and Hearing Protection |
|---|---|---|---|---|---|
| 6 Trigger | 6.1 command error | 5 | 5 | 25 | Design trigger for usability. Include safety interlock |
| | 6.2 ergonomics | 4 | 3 | 12 | Design padding into handle. Ibid (use gloves) |
| 7 Battery Power supply | 7.1 fire | 5 | 3 | 15 | Design thermal cut-out into battery electronics |
| | 7.2 heat | 3 | 6 | 18 | Design thermal cut-out into battery electronics |
| | 7.3 electric shock | 8 | 2 | 16 | Design tool to operate on less than 30VDC |
| | 7.4 reverse polarity | 2 | 3 | 6 | Design electronics so that assembly errors are physically prevented. |
| 8 Charging station | 8.1 fire | 8 | 5 | 30 | Design thermal cut-out into charger electronics |
| | 8.2 electric shock | 8 | 4 | 24 | Design double insulated charging station, no external metal parts that are easily accessible, charging pins to be difficult to reach even accidentally |
| 9 Disk changing tool | 9.1 left attached and flung off | 5 | 3 | 15 | Design tool to be loose fit so it will not stay attached. |
| | 9.2 lost so disk not attached properly | 4 | 5 | 20 | Design tool out altogether. Alternatively provide a means to retain it with the tool. |

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
