# Peer review of "Alignment of the Safety Assessment Method with New Zealand Legislative Responsibilities"

_safety, 2019_

Round 1

Reviewer 1 Report

The author seeks to propose an improved and systematic approach to assess safety hazards and promote safety management efforts. The presented work can be usefully adopted in workplaces to reduce the likelihood of risk. However, the presentation of the ideas can be significantly improved to enhance readability. Below are a few additional comments that can be addressed:

1) The introduction does not sufficiently capture the problem that the authors seek to address or the contributions of the study. The section simply presents a number of definitions which appears to be very disjoint from the objectives of the article. The introduction can be improved by clearly presenting the purpose of the article.

2)In the background section, the authors present some limitations of existing hazard assessment approaches. However, for the most part, no literature is cited and the author has not sufficiently demonstrated to readers that these limitations exist. It is suggested that the author explicitly identify existing approaches with relevant citations while pointing to the limitations.

3) The background section can significantly be improved to enhance readability. For example, the authors mention that some scales fail to recognize the non-linear effect of catastrophic accidents. It is really not clear what the authors mean until we get to the subsequent sections of the article. Likewise, the author mentions that current scales bear no resemblance to the definitions of H&S legislation. The article can significantly be improved if the authors provide more detailed context for the claims that are made along with representative examples.

4) There is a typo in line 83.

5) What does hazard progression mean in line 160? It is suggested that the author gives a definition and example of a case or incident for each of the harm scale items. This will clarify how the scale can be used.

Author Response

Reviewer 01

Author's response

1) The introduction does not sufficiently capture the problem that the authors seek to address or the contributions of the study. The section simply presents a number of definitions which appears to be very disjoint from the objectives of the article. The introduction can be improved by clearly presenting the purpose of the article.

The purpose statement has now been included in the Introduction. Also, Section 2  has been extensively re-written, and should address some of these concerns as it includes  a better description of the context, and an improved rationale.

2)In the background section, the authors present some limitations of existing hazard assessment approaches. However, for the most part, no literature is cited and the author has not sufficiently demonstrated to readers that these limitations exist. It is suggested that the author explicitly identify existing approaches with relevant citations while pointing to the limitations.

The literature review component has been re-written and substantially enlarged.

3) The background section can significantly be improved to enhance readability. For example, the authors mention that some scales fail to recognize the non-linear effect of catastrophic accidents. It is really not clear what the authors mean until we get to the subsequent sections of the article. Likewise, the author mentions that current scales bear no resemblance to the definitions of H&S legislation. The article can significantly be improved if the authors provide more detailed context for the claims that are made along with representative examples.

The section on non-linearity of scales has been expanded, and some examples given.

4) There is a typo in line 83.

Thank you, this is now corrected

5) What does hazard progression mean in line 160? It is suggested that the author gives a definition and example of a case or incident for each of the harm scale items. This will clarify how the scale can be used.

This simply means that the hazard is controlled by existing treatments.  This clarification has been added.

Reviewer 2 Report

The paper titled “Alignment of the safety assessment method with legislative responsibilities” deals with the need to adjust the safety assessment process to the developments in the health and safety legislation. The paper uses New Zealand as an example, which I believe should be put in the title of the paper.

This is not your typical research paper, and is written in a much more “free style” which makes it difficult to follow and comment on many aspects which are usually necessary in a research paper. The abstract should be more specific regarding the topic at hand. Currently, it is too general.

The Introduction is relatively short and only 3 references are cited. In general only 5 references are cited throughout the Manuscript, which shows that the Authors are offering their own opinion on a matter, not that they are considering options which exist worldwide, develop and test a methodology (again, no research). This, by no means, decreases the value of the paper, but requires attention.

The Author starts with a non-standard definition of hazard assessment. Maybe the definition he provides is more applied to risk assessment, making the first paragraph of the Introduction unclear (or wrong?). Hazard is a substance, agent, event which by itself can cause harm. Risk assessment deals with how big is the harm, how probable it is, and who would be affected. The Author provides a general Introduction to hazard assessment, although it is difficult to follow the point, having in mind the title of the paper.

To someone not familiar with the system in NZ, this part of the Manuscript “Background to health and safety legislation in NZ” is quite difficult to read. The Authors could at least try to bring the problems they are attempting to solve closer to the European legislation or any other system.

Line 55: what is “(1.5)”?

The methodology is too vaguely described, and, as the rest of the paper, leaves the reader with the sensation that the Author is expressing his own opinion and attitudes on the matter, offering some solutions, without any way to verify if this is ok, better, or the best possible way to do.

Line 157: items

Line 177: name the source, not only the number of the reference.

The Author states that the work is conceptual in nature, therefore lacking verification. In general, I find the paper interesting and even meriting publication, although I believe the Authors could make an effort to shorten the paper, provide more examples, reorganize the paper so it is easier to read (maybe moving the large tables to supplementary material) and maybe demonstrating on a simple example how their method differs from something already in use.

Author Response

Reviewer 02

The paper titled “Alignment of the safety assessment method with legislative responsibilities” deals with the need to adjust the safety assessment process to the developments in the health and safety legislation. The paper uses New Zealand as an example, which I believe should be put in the title of the paper.

Agreed, change made to ' Alignment of the safety assessment method with New Zealand legislative responsibilities  '

This is not your typical research paper, and is written in a much more “free style” which makes it difficult to follow and comment on many aspects which are usually necessary in a research paper. The abstract should be more specific regarding the topic at hand. Currently, it is too general.

The paper has received major edits, including to the literature review, rationale for the need, and method.

The abstract has been rewritten.

The Introduction is relatively short and only 3 references are cited. In general only 5 references are cited throughout the Manuscript, which shows that the Authors are offering their own opinion on a matter, not that they are considering options which exist worldwide, develop and test a methodology (again, no research). This, by no means, decreases the value of the paper, but requires attention.

The literature review has been expanded to provide a better context and rationale for the work.  

The Author starts with a non-standard definition of hazard assessment. Maybe the definition he provides is more applied to risk assessment, making the first paragraph of the Introduction unclear (or wrong?). Hazard is a substance, agent, event which by itself can cause harm. Risk assessment deals with how big is the harm, how probable it is, and who would be affected. The Author provides a general Introduction to hazard assessment, although it is difficult to follow the point, having in mind the title of the paper.

Indeed there are many uses of the term ‘hazard’. I see your point, and have edited the opening statement to more clearly identify that the context is hazard as understood in occupational health and safety.

To someone not familiar with the system in NZ, this part of the Manuscript “Background to health and safety legislation in NZ” is quite difficult to read. The Authors could at least try to bring the problems they are attempting to solve closer to the European legislation or any other system.

This section has received substantial attention, and I have tried to better relate the NZ system to the wider literature.

Line 55: what is “(1.5)”?

(typically ranging from 1 to 5). Manuscript has been edited.

The methodology is too vaguely described, and, as the rest of the paper, leaves the reader with the sensation that the Author is expressing his own opinion and attitudes on the matter, offering some solutions, without any way to verify if this is ok, better, or the best possible way to do.

The manuscript now has a more explicit purpose statement added, which is proceeded by a rationale or description of the need based on the literature.  The work is a conceptual contribution, and I hope the edits have made this clearer.

Line 157: items

Thank you, this is now corrected

Line 177: name the source, not only the number of the reference.

Source is now named in the text.

The Author states that the work is conceptual in nature, therefore lacking verification. In general, I find the paper interesting and even meriting publication, although I believe the Authors could make an effort to shorten the paper, provide more examples, reorganize the paper so it is easier to read (maybe moving the large tables to supplementary material) and maybe demonstrating on a simple example how their method differs from something already in use.

The example scenario has been moved to an appendix to shorten the paper.

I feel the other tables are essential to the preservation of the concept, and would prefer to leave them in place.

Reviewer 3 Report

This paper is about an interesting and relevant topic. The authors proposed a new risk assessment method, supported in NZ legislation. However, I found several weaknesses in this paper.

The study objective should be clarified in the introduction section. The research problem was also not clear for me in the first time that I read the introduction section.

A strong literature review to support the methodology proposed is missing.

Several concepts need to be reviewed along the text. For example: If the author is using ISO 31000 as a reference, the terms should be aligned. E.g. Hazard assessment should be replaced by risk assessment. Lines 53 and 54 – If the severity and probability are determined qualitatively, the method is qualitative and not semiquantitative. It was also not clear what the author understand by quantitative method. A quantitative risk assessment method is when probability and consequence are determined quantitatively. In table

I do not agreed with several statements made by the author. The authors are proposing a risk assessment method. This is legitime. However, asking that the same method be applied in all organizations is not correct in my point of view. In Line 61 the authors lead the reader to realize that the use of different scales is wrong. But this is not true. The scales and acceptance criteria can be adjusted to each organization reality. For example, a risk can be considered acceptable in a construction company and unacceptable in a supermarket. Additionally, even in the same sector what is acceptable or not depends of the organization and their safety culture level. Acceptance criteria need to become stringent as the safety culture level of the organization increase. The authors also used the proposed method to assess the risk of accidents and occupational diseases. Workers exposure to physical, chemical and microbiological agents should be assessed quantitatively, using appropriate equipment and methodologies and compared to quantitative acceptance criteria defined in legislation or guidelines.

The risk assessment methodology raises some doubts: Lines 159-166 –The first consequence level is “Hazard present but existing controls prevent progression”. This is not a consequence. I suggest to start with Level 2. Level 4 is difficult to understand. How the acceptance criteria were defined? 

Table 4: The authors present the hazard. But the likelihood and the consequence is for the corresponding risk and not for the hazard.

Author Response

Reviewer 03

This paper is about an interesting and relevant topic. The authors proposed a new risk assessment method, supported in NZ legislation. However, I found several weaknesses in this paper.

The study objective should be clarified in the introduction section. The research problem was also not clear for me in the first time that I read the introduction section.

The objective has now been made clearer in the instruction, a rationale is provided in the literature review, and a detailed purpose statement in the method.

A strong literature review to support the methodology proposed is missing.

The literature review has now received substantial edits.

Several concepts need to be reviewed along the text. For example: If the author is using ISO 31000 as a reference, the terms should be aligned. E.g. Hazard assessment should be replaced by risk assessment. Lines 53 and 54 – If the severity and probability are determined qualitatively, the method is qualitative and not semiquantitative. It was also not clear what the author understand by quantitative method. A quantitative risk assessment method is when probability and consequence are determined quantitatively. In table

The introduction has been changed to make it clear that hazards are to be understood in the context of occupational health and safety.

I would prefer to keep the term semi quantitative, because the process does result in an overall risk number, and this is material to the decision making about which risks to treat. I agree this is not really quantitative, but it does the same thing. Furthermore, if one looks at a quantitative method like FMEA, they too are using subjective assessments of probabilities.

I have added further content about the ordinal scales to make the weaknesses of this approach clear.

I do not agreed with several statements made by the author. The authors are proposing a risk assessment method. This is legitime. However, asking that the same method be applied in all organizations is not correct in my point of view. In Line 61 the authors lead the reader to realize that the use of different scales is wrong. But this is not true. The scales and acceptance criteria can be adjusted to each organization reality. For example, a risk can be considered acceptable in a construction company and unacceptable in a supermarket. Additionally, even in the same sector what is acceptable or not depends of the organization and their safety culture level. Acceptance criteria need to become stringent as the safety culture level of the organization increase. The authors also used the proposed method to assess the risk of accidents and occupational diseases. Workers exposure to physical, chemical and microbiological agents should be assessed quantitatively, using appropriate equipment and methodologies and compared to quantitative acceptance criteria defined in legislation or guidelines.

I think there may be a misunderstanding here. The purpose of the work is to provide a standardised instrument for risk assessment that can be used in the absence of any other more robust method, and instead of the ad hoc constructs that organisations create in their attempt to meet the non-specific requirements of the legislative frameworks. While the results given in this paper do not meet all the desirably attributes expressed in the purpose statement, they do perhaps move the field forward towards a possible future more standardised risk assessment instrument for H&S. Movement has been made towards a somewhat standardised consequence that embodies the concepts of organisational risk appetite. It also provides a more explicit prompt for communication within the organisation than is achieved by methods currently in use.  The resulting method more explicitly prompts for recovery mechanisms (in addition to preventative actions) for catastrophic risks. In putting this method forward the paper does not preclude the use of other methods. As a whole this particular risk assessment approach may have merit for use by organisations that desire to improve their management of H&S risks but are small, in developing nations, or in other ways lack resources to develop their own methods.

The risk assessment methodology raises some doubts: Lines 159-166 –The first consequence level is “Hazard present but existing controls prevent progression”. This is not a consequence. I suggest to start with Level 2. Level 4 is difficult to understand. How the acceptance criteria were defined? 

The harm scale needs a zero  point, which is common is other scales too.

In this case the intended meaning is that the hazard is controlled by existing treatments and if it materialises the effects are expected to be inconsequential. Text to this effect has been added to the paper.

Level 4 is adapted directly from the NZ Health and Safety at Work Act (2015), which may be why the terms are unfamiliar.

The colour bands in Figure 1 are the same as those in Table 2, and all the content thereof is simply a proof of concept of how the alignment and harmonisation can be achieved. The decision thresholds have been determined subjectively. If this system was to be implemented in an organisation it is recommended that Table 2 be adapted as needed, and then Figure 1 changed accordingly.

Text to this effect has been added to the paper.

Table 4: The authors present the hazard. But the likelihood and the consequence is for the corresponding risk and not for the hazard.

Yes this is always a moot point in H&S risk assessment.  Fault tree analysis has a much better representation of root causes and the hazards that they cause. However in all H&S risk assessments I have seen, the whole fault tree is effectively collapsed into one risk or hazard.

Reviewer 4 Report

The limits of the study are presented. In the example, residual risk is not estimated. The tool is not validated (tested). Different participants have not used the tool. Subjectivty has not been assessed. The paper is well written. However there are many papers (Safety Science, Risk Analysis etc. ) which deal with risk estimation. Some papers are specific to risk estimation for machinery. The manuscript will be enhanced if the results from those papers are discussed and compared to the proposed tool. Thank you.

Author Response

 Reviewer 04

The limits of the study are presented. In the example, residual risk is not estimated. The tool is not validated (tested). Different participants have not used the tool. Subjectivty has not been assessed. The paper is well written

The residual risk after treatment has not been shown here, but the principles are evident in Table 3.

Lack of validation has been identified as a limitation, and has been expanded on in the limitations section.

However there are many papers (Safety Science, Risk Analysis etc. ) which deal with risk estimation. Some papers are specific to risk estimation for machinery. The manuscript will be enhanced if the results from those papers are discussed and compared to the proposed tool.

The literature review has been expanded to include more papers. In doing so I have tried to focus on the national policy development, and the link to risk assessment practices.  

Round 2

Reviewer 1 Report

The article makes a useful contribution.

Reviewer 2 Report

The Authors have made significant changes to the Manuscript which is now clearer and offers value to the reader. I congratulate the Authors on their effort.

Reviewer 4 Report

The manuscript has been improved.